# Point singularity array with metasurfaces

**Soon Wei Daniel Lim** [1,3] ✉, **Joon-Suh Park**[1,2,3], **Dmitry Kazakov** [1], **Christina M. Spägele**[1], **Ahmed H. Dorrah**[1], **Maryna L. Meretska** [1] & **Federico Capasso** [1]

Phase singularities are loci of darkness surrounded by monochromatic light in a scalar field, with applications in optical trapping, super-resolution imaging, and structured light-matter interactions. Although 1D singular structures, like optical vortices, are common due to their robust topological properties, uncommon 0D (point) and 2D (sheet) singularities can be generated by wavefront-shaping devices like metasurfaces. With the design flexibility of metasurfaces, we deterministically position ten identical point singularities using a single illumination source. The phasefront is inverse-designed using phase-gradient maximization with an automatically-differentiable propagator and produces tight longitudinal intensity confinement. The array is experimentally realized with a $TiO_2$ metasurface. One possible application is blue-detuned neutral atom trap arrays, for which this field would enforce 3D confinement and a potential depth around 0.22 mK per watt of incident laser power. We show that metasurface-enabled point singularity engineering may significantly simplify and miniaturize the optical architecture for super-resolution microscopes and dark traps.

Optical singularities occur when some parameter of the electric field is undefined; for instance, phase singularities occur when the wavefront phase is undefined at field zeros, and polarization singularities occur when at least one parameter of the polarization ellipse is undefined[1]. For random monochromatic scalar fields in a 3D space, such as in speckle patterns, 1D linear singularities (lines or curves) are ubiquitous since they are robust against field perturbations. On the other hand, 0D (point) and 2D (sheet) singularities are far less common as they do not share the same robustness. They tend to fragment into stable 1D linear singularities upon field perturbation[2], such as stray light either originating from external sources or deviations from the desired geometrical parameters of optical devices. Nevertheless, 2D singularities (membranes of darkness in 3D space) have been engineered and experimentally realized using wavefront shaping devices like metasurfaces[3]. Such devices can be obtained by inverse design optimization so that the light field achieves a large spatial gradient of the phase normal to the surface comprising the singularity.

While it is straightforward to position bright spots of light using conventional computer generated holography techniques such as

Gerchberg-Saxton phase retrieval[4,5], these methods perform poorly at structuring dark regions of subwavelength dimensions[3]. 0D point singularities require the scalar field to vanish at only one point. These cold spots have been identified in the near-field of nanoparticles[6] and individual spots may be controllably displaced by superposing plane waves[7]. Lattices of points with vanishing intensity in a vector polarization field have also been generated in the transverse plane[8,9]. Here, we seek a method for deterministically placing multiple 0D singularities that is not bound to periodic spacing and does not mandate the use of multiple beams.

We present a straightforward method of deterministically positioning point singularities in a cylindrically-symmetric field. This strategy produces singularities with tight confinement, i.e., small characteristic spatial dimensions with a rapid increase of the field intensity (amplitude modulus squared) away from the singularity point. We begin by describing the physical intuition behind the phase gradient maximization technique based on the geometrical structure of the 0D singularity. While the singularities are engineered for a scalar field corresponding to a fixed linear polarization, we also examine the

[1]Harvard John A. Paulson School of Engineering and Applied Sciences, 9 Oxford Street, Cambridge, MA 02138, USA. [2]Nanophotonics Research Center, Korea Institute of Science and Technology, Seoul 02792, Republic of Korea. [3]These authors contributed equally: Soon Wei Daniel Lim, Joon-Suh Park ✉ e-mail: lim982@g.harvard.edu

full 3D polarization distribution that would be generated by a realistic wavefront shaping device like a metasurface. We then experimentally realize a linear array of ten tightly confined point singularities in the axial direction with a metasurface comprising $TiO_2$ nanopillars on glass. As a potential application, we evaluate the suitability of the resultant singular fields for neutral atom trapping in the blue-detuned regime, in which atoms are trapped in positions of darkness. While the engineered singularity array is very sensitive to the tilt of incident illumination, it is robust to wavelength changes of the trapping laser and demonstrates 3D confinement with no escape channels. Metasurface-enabled traps have the potential to greatly simplify the optical architecture required to produce dark optical traps for atoms or larger particles.

## Results

### Geometry and topology of 0D singularities

Point, i.e., 0D, phase singularities occur when a complex scalar field $E$ is zero at only one point. That is, the real and imaginary zero-isosurfaces of $E$, loci of points for which $Re(E) = 0$ and $Im(E) = 0$, respectively, touch tangentially at only one point (Fig. 1a). The field phase is defined for every point around the singularity except for the point itself (Fig. 1b) and the intensity decreases quadratically to zero towards the singular point (Fig. 1c). Similar to 2D sheet-like singularities, 0D point singularities are uncommon and fragile. This is because they lack a property known as topological protection and hence occur rarely in nature[3,10]. Nevertheless, they can be engineered to closely approximate 0D singularity behavior to within measurement uncertainties.

Topological protection refers to the robustness of a system against perturbations or defects, provided certain topological characteristics are preserved. For example, symmetry-protected bound states in the continuum (BICs) are robust and continue to exist under small changes in system parameters, because of conserved and quantized topological charges that can only change under large system parameter deviations[11]. In singular optics, topological protection refers to the persistent existence of the singularity under small changes in the surrounding medium or wave properties. They are referred to as elementary optical singularities and include, for instance, the canonical orbital-angular-momentum (OAM) singularity in a scalar field or bright C-points in polarization fields[12]. Upon perturbation, such elementary optical singularities are merely displaced and can only be annihilated by a large perturbation that merges two singularities of opposite charge. Singularities that are not topologically protected, such as higher-order OAM modes, do not have this guarantee; they are destroyed or split into multiple topologically-protected singularities under field perturbation. This is also observed in BICs when symmetry-breaking operations cause a V-point to fragment into topologically-protected C-points[13]. This fragility to perturbation renders such singularities uncommon in nature[10].

### Engineering singularity confinement

As with singularities of other geometries, the 0D singularity is accompanied by a region of large phase gradient magnitude $|\nabla\phi|^2 = (\partial_x\phi)^2 + (\partial_y\phi)^2 + (\partial_z\phi)^2$ (Fig. 1d) which diverges to infinity at the position of the singularity. In optical fields, this phase accumulation rate can be much larger than the field wavenumber $k = 2\pi/\lambda$, indicating superoscillatory behavior[14].

While phase singularities can be engineered by enforcing perfect destructive interference at a point, the confinement of the dark point is

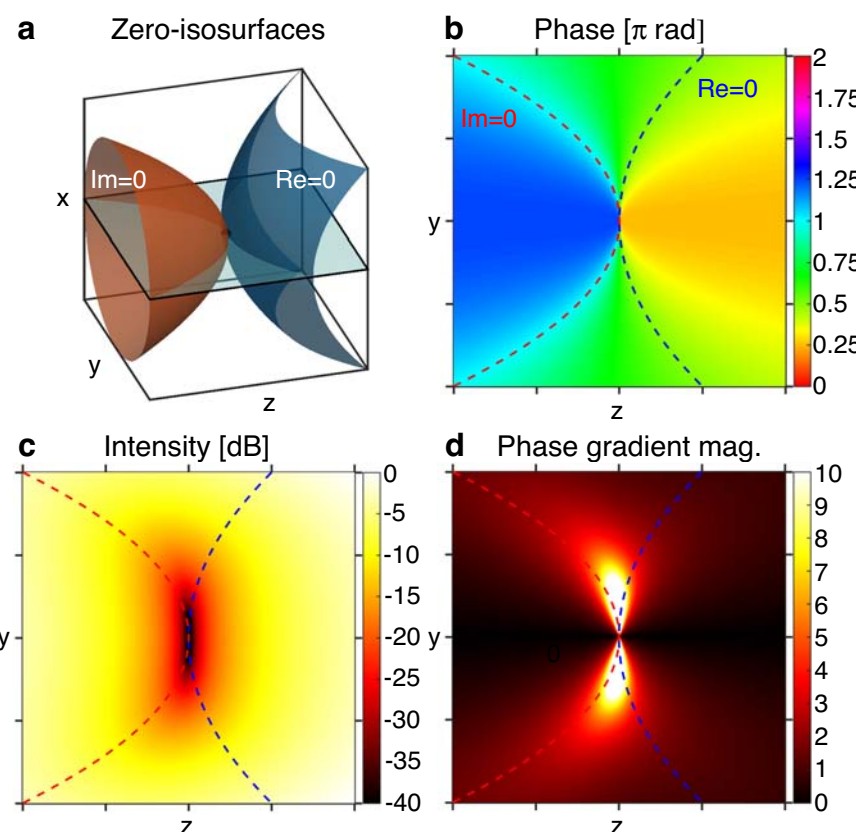

**Fig. 1 | 0D singularity geometry. a** 0D singularities in 3D space are isolated points of vanishing intensity in a scalar field $E$, occurring when the real (blue) and imaginary (red) zero-isosurfaces of $E$ intersect tangentially. **b** $yz$ cross-sectional phase and (**c**) intensity profiles of the 0D singularity in (**a**). The dotted blue and red lines represent the real and imaginary zero-isolines of $E$ on the plane, respectively. **d** Magnitude of the phase gradient $|\nabla\phi|$ in the $yz$ plane, which is dominated by the minus $z$-directed phase gradient. The phase gradient diverges to infinity at the singularity position.

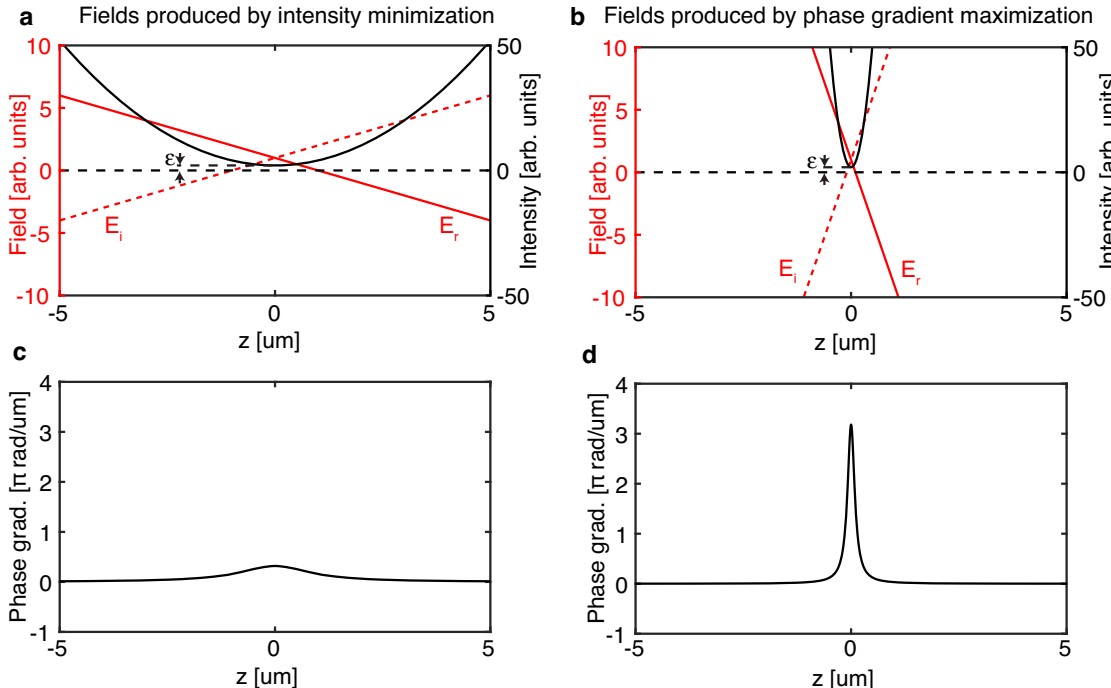

**Fig. 2 | Comparison between two methods of producing 0D singularities: intensity minimization and phase gradient maximization.** Only field behavior along the optic axis ($z$ axis) is shown for simplicity. **a** Real ($E_r$) and imaginary ($E_i$) parts of scalar field $E$ in the vicinity of a low intensity position with minimum intensity $\epsilon$. Intensity minimization at $z = 0$ does not take the spatial distribution of fields around the low intensity point into account, producing fields with slowly varying $E_r$ and $E_i$ through the minimum, thereby producing a broad intensity minimum. **b** On the contrary, since phase gradient maximization at $z = 0$ simultaneously minimizes the intensity there and maximizes the field slopes $\frac{dE_r}{dz}$, $\frac{dE_i}{dz}$ passing through that point, the resultant intensity minimum is narrow. **c** The phase gradient peak through $z = 0$ for the field in (**a**) produced by intensity minimization there is typically much lower than that of phase gradient maximization, as depicted in (**d**), which plots the phase gradient for the field in (**b**).

another critical parameter, especially in superresolution microscopy (e.g., STED[15]) and optical trapping. In these applications, dark positions should ideally be fully surrounded by light (i.e., 3D confinement) with sharp field gradients (i.e., tightly confined/localized). These additional constraints on the field distribution in the vicinity of a dark point cannot be satisfied by simply minimizing the field intensity at the target position of the 0D singularity. Here, we show that phase gradient maximization can enforce singular behavior at a point while simultaneously achieving tight confinement around the singularity. To build intuition for this technique, we first consider a complex field $E$ along a line, and compare the fields that are produced by a simple intensity minimization at $z = 0$ and a phase gradient maximization at that same point (Fig. 2). To avoid plotting unrealistically high phase gradients, we show fields that have a finite minimum intensity $\epsilon > 0$. Such a system may not yield zero intensity due to fabrication imperfections or optimization constraints. Optimization constraints arise when one seeks to balance multiple competing desired behaviors, e.g., in a multi-objective optimization for which one simultaneously optimizes the field structure at different locations. Close to an intensity minimum, the real and imaginary field components ($E_r$ and $E_i$, respectively) are approximately linear (Fig. 2a). Since engineering the singularity by minimizing the field intensity at $z = 0$ just enforces a small $\epsilon$ there, it is insensitive to the slopes of $E_r$ and $E_i$ across the singularity, which can be shallow and thus produce a slowly varying field intensity minimum with weak localization. At $z = 0$, $E_r$ and $E_i$ change sign and thereby produce a $\pi$ phase shift across the field minimum. This shift in phase can be captured by the variation in phase gradient $\partial_z \phi$, which has a broad and short peak at $z = 0$ (Fig. 2c). Engineering a singularity by simply minimizing the intensity at the desired field minimum position does not give one control over the confinement there.

On the other hand, maximizing the phase gradient at $z = 0$ simultaneously achieves singular behavior and improves confinement.

Intuitively, noting that the phase gradient can be written in terms of field gradients $\nabla \phi = \text{Im}(\nabla E / E)$, maximizing $\nabla \phi$ not only minimizes the value of $E$ in the denominator but also maximizes the field gradients $\nabla E$ in the numerator. This means that the slopes of $E_r$ and $E_i$ are steeper across the singularity, producing a more rapidly varying field intensity minimum with narrower spatial confinement (Fig. 2b). A higher peak phase gradient also yields a taller and narrower phase gradient peak across the field minimum so that the accumulated phase across the minimum remains $\pi$ (Fig. 2d).

In three dimensions, 0D singularities are characterized by large phase gradients in all directions. One has to simultaneously maximize the phase gradients at the same point to squeeze the singularity into a point, a task which poses convergence difficulties since changing one directional gradient at a point inevitably affects the other gradients in the other directions. This problem is circumvented when the field is constructed to be azimuthally (cylindrically) symmetric about the optical axis: i.e., the electric field $\mathbf{E}(r,z)$ is only a function of the radial distance from the optical axis $r$ and longitudinal position along the optical axis $z$. One can produce 0D point singularities along the optical axis just by maximizing one directional gradient at each of the desired points: the $z$-directed phase gradient. This exploitation of a system symmetry improves numerical convergence to an optimal design.

## Optimization approach for metasurface realization

As a proof-of-concept for 0D singularity engineering, we designed an array of ten 0D singularities spaced 3 μm apart (Fig. 3a) to be generated by a phase-only metasurface measuring 1 mm in diameter, and illuminated by a narrowband laser centered at $\lambda = 760$ nm. Although we demonstrate a uniform array of singularities here, the algorithm can be applied to aperiodic singularity patterns as well, and we show one such design in Supplementary Figure 1. Such a light field, structured longitudinally along the optical axis, is challenging to generate

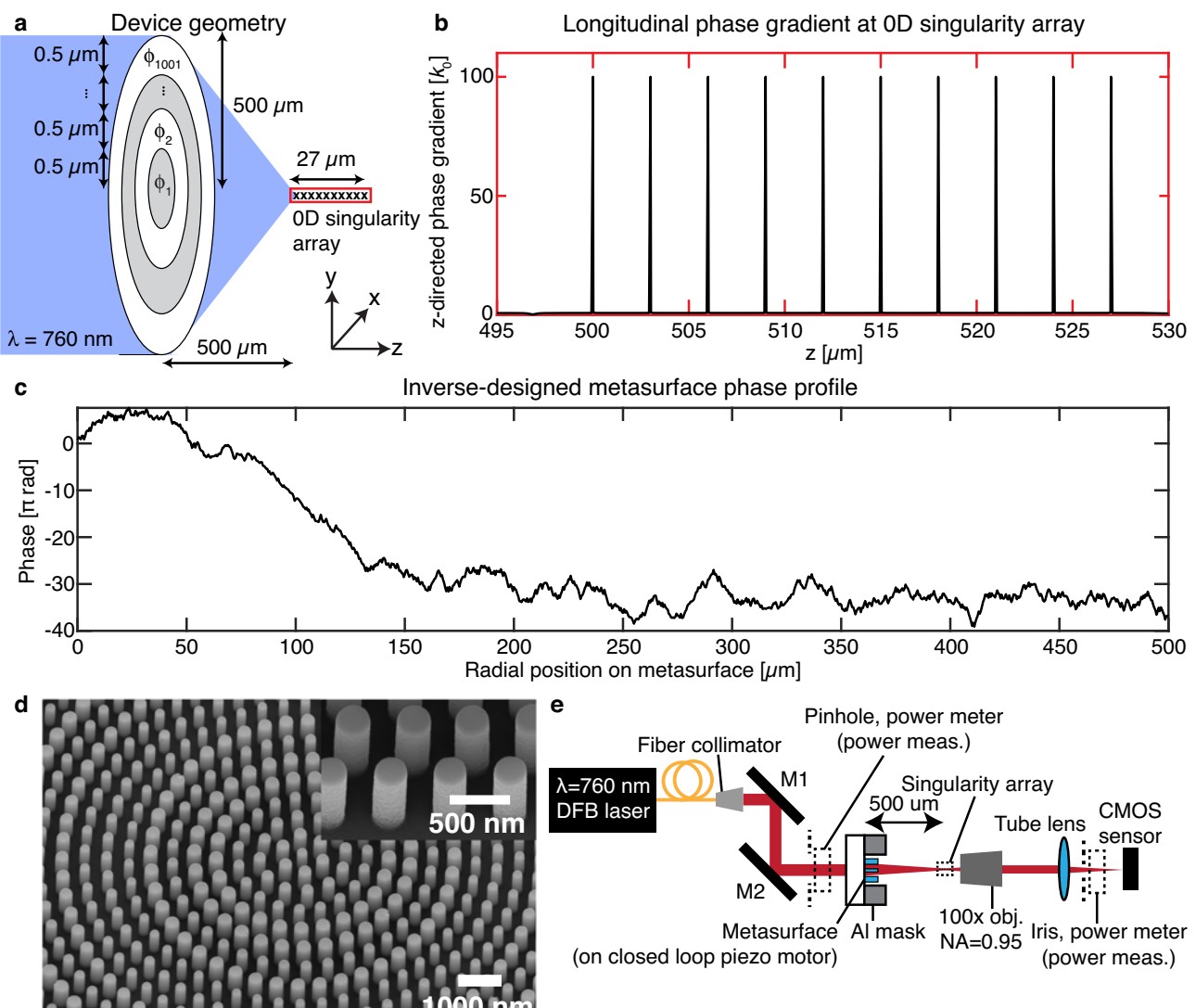

**Fig. 3 | Design and experimental realization of 0D singularity array. a** Geometry of the phase-only metasurface to generate the singularity array upon illumination by λ = 760 nm light. The Cartesian directions are also indicated. **b** Longitudinal ($z$) phase gradient along the optic axis at the 0D singularity array, demonstrating large (compared to the free-space wavenumber $k_0$) and uniform phase gradients at the singularity locations. **c** Inverse-designed metasurface phase profile as a function of metasurface radial position that achieves the 0D singularity array. The phases have been unwrapped to show the long-range variation. **d** Scanning electron microscope image of the $TiO_2$ nanopillars on $SiO_2$ at the center of the fabricated metasurface that achieves the phase profile in (**c**). Inset: close-up of the nanopillars demonstrate vertical sidewalls. **e** Experimental setup to generate and characterize the 0D singularity array. Dotted lines indicate the positions of the pinholes and power meter used in characterizing the absolute transmission intensity.

using conventional holography methods that excel at designing only transverse field patterns. Full 3D holographic pattern generation with both transverse and longitudinal control remains an area of active research[16]. We partition the cylindrically symmetric metasurface plane into 1001 annular regions, each 500 nm thick. Each annular region is assigned a transmission phase delay so that the metasurface system can be parametrized by the 1001 phase delay values which serve as tunable optimization parameters. The phase profile from the metasurface is propagated into free space ($z > 0$) using a vectorial propagator[17] built on an automatically differentiable platform (Tensorflow[18]), assuming that the incident field is linearly $x$-polarized for simplicity. The process is generalizable to optimizing both transverse polarization components over the surface and is not restricted to single scalar fields. This automatically differentiable propagator affords computationally efficient calculation of the exact numerical gradients of arbitrary objective functionals on the diffracted field.

There are two steps in the optimization process. In the first stage of optimization, we maximize the longitudinal phase gradient of the $x$-polarized $E_x$ field at ten equally-spaced target singularity positions from $z = 500\,\mu m$ to $z = 527\,\mu m$ along the optical axis. The radially oriented phase gradient is identically zero due to azimuthal symmetry and continuity conditions for analytic fields: a nonzero radial phase gradient along the optical axis will produce a kink in the phase gradient across the optical axis. This first step produces a 0D singularity at each of the target positions. The intensity (i.e., $|E_x|^2 + |E_y|^2 + |E_z|^2$) and $E_x$ phase profiles around each of the singular positions after this first step are plotted in Supplementary Figures 2 and 3, respectively. In several positions, the real and imaginary zero-isolines come close but do not touch, indicating that these situations are close approximations of 0D singularities and not mathematical 0D singularities. In the second stage of optimization, we use the optimized first stage result to equalize the phase gradient and second spatial derivative of $|E_x|^2$ (as a proxy for the intensity) over all the singularity positions and thus

obtain nearly identical singularities across the array. The field intensity and phase profiles around each dark position are plotted in Supplementary Figs. 4–5, respectively. The phase gradient profile of the $E_x$ field along the optical axis is plotted in Fig. 3b and shows identical large superoscillatory values of $100k_O$ at the singularity positions, as designed. High spatial resolution plots of the phase gradients around each singularity position are shown in Supplementary Fig. 6, which also show that the full-width-at-half-maximum of the phase gradient magnitude is 2.3 nm for each singularity. The tight feature localization of optical singularities has been exploited for precision displacement sensing[19]. The complex electric field components $E_{x,y,z}$ on the transverse plane under x-polarized illumination at the metasurface plane at each singularity position are plotted in Supplementary Fig. 7 and the spatially-varying transverse polarization states (parametrized by the polarization ellipse distribution) are plotted in Supplementary Fig. 8. The cross-polarized electric fields $E_{y,z}$ are generated from the vectorial propagation of the x-polarized field after the metasurface plane. $E_{y,z}$ vanishes on-axis due to cylindrical symmetry and the fields close to the optical axis are predominantly x-polarized. An isolated zero field intensity position in a linearly polarized field is known as a V-point polarization singularity[20,21], thus if the metasurface is illuminated with linear polarized light, it produces an array of V-points. The inverse-designed phase profile along the metasurface is unwrapped and plotted in Fig. 3c to show the long-range structure. Full details of the optimization process are in Supplementary Information section 1.

The azimuthal symmetry of the wavefront-shaping metasurface results in negligible OAM about the optical axis. Although singular optics and OAM are frequently discussed in the same context, they are distinct concepts in complex beams comprising superpositions of many OAM eigenmodes[22]. The transverse OAM density about the optical axis and transverse components of the Poynting vector for each of the 0D singularities (under x-polarized illumination) is plotted in Supplementary Fig. 9, which demonstrate negligible OAM and no energy circulation about the optical axis.

Although optical singularities are also located close to large intensity gradients, maximizing the z-directed intensity gradient instead of the phase gradient as a proxy for producing such singularities does not afford precise control over the position of minimum intensity. The resultant positions of minimum intensity also do not have phase gradients appreciably larger than the vacuum wavenumber. Supplementary Information section 2 discusses the result of optimizing the intensity gradients for a system of the same geometry and the resultant field profiles are plotted in Supplementary Fig. 10.

We fabricated a 1 mm diameter metasurface comprising 700 nm tall cylindrical $TiO_2$ pillars on a fused silica substrate to enforce the required phase profile and generate the ten 0D singularities. The fabrication process is similar to previously published work[23] and involves electron beam lithography of the required nanopillar profile into electron beam resist, followed by atomic layer deposition of amorphous $TiO_2$ into the developed resist voids. Over-deposited $TiO_2$ is etched back using reactive ion etching to leave free-standing nanopillars. An opaque aluminum aperture is positioned around the metasurface to reduce stray light. Details of the nanofabrication process are in the Methods section and the nanopillar library optical performance is plotted in Supplementary Fig. 11. At each metasurface position, we pick the nanopillar from the library that has the closest transmitted phase to the required phase at that radial position. The non-uniform transmission amplitude of the meta-atom library introduces slight field deviations from the design field distribution, and we plot the predicted field intensity and phase profiles incorporating these imperfections in Supplementary Figs. 12, 13, respectively. The field intensity structure is largely preserved but the phase profile is slightly distorted near the intended singularity positions. This deviation arises because the 0D singularities are not topologically protected and were constructed by finely-tuning the metasurface phase profile

under the assumption of ideal uniform transmission. Non-idealities arising from a realistic nanopillar library thus slightly distort the dark regions so that these positions are not mathematical singularities. While this deviation is likely not significant enough to impact applications which are sensitive to intensity profiles, closer field behavior to the mathematical ideal can be obtained by including nonuniform transmission intensity of the nanopillar library during optimization[24]. Scanning electron microscope images of the fabricated metasurface are shown in Fig. 3d. For characterization, the metasurface is illuminated with a narrowband distributed feedback diode laser (λ = 760.9 nm, 2 MHz linewidth) coupled to a single mode fiber with collimated output, and the transmitted field through the metasurface is captured over 1201 longitudinal z-positions at steps of 50 nm, where z = 0 mm corresponds to the patterned surface of the metasurface, using a high magnification objective (×100, NA = 0.95) in a horizontal microscope system (Fig. 3e). The transmitted intensity measurements are normalized to the incident power flux at the metasurface. Full experimental and data processing details are in Supplementary Information section 3.

## Experimental longitudinal field profile

The simulated cylindrically symmetric field intensity profile on the xz plane in the vicinity of the ten 0D singularities is plotted in Fig. 4a. The experimental intensity profiles in the longitudinal xz and yz planes are displayed in Fig. 4b, c, respectively, and demonstrate good agreement with the simulated profiles. The intensity profile colormaps are adjusted to show the singularity region clearly and some parts of the surface plots are intentionally saturated to better show the singular regions. The unsaturated intensity profiles are plotted in Supplementary Fig. 14. The maximum intensity value is indicated adjacent to each plot. The on-axis intensity comparison between the numerical and experimental cuts is plotted in Fig. 4d. The longitudinal cuts were obtained by stacking the 1201 captures of the transverse field intensity. The captured transverse xy field intensity at and between the ten singular positions are shown in Fig. 4e, with rings of light around the dark singular points and bright on-axis spots in between singular positions. These transverse intensity pictures are stacked in the longitudinal direction to produce the xz and yz cuts in Fig. 4b, c, respectively. The axial displacement of the experimental intensity profile with respect to the simulated profile in the negative z-direction can be attributed to the incident laser wavelength of 760.9 nm being slightly longer than the design wavelength of 760 nm.

We observe that the experimental intensity is about a factor of four times smaller than the numerically predicted intensity. This is due to our intensity normalization choice and diffractive losses from the breaking of the ideal periodic boundary condition that underlies our metasurface library. We underestimate the field intensity by measuring the transmitted field power profile after it passes through the microscope objective and tube lens, thereby incorporating the reflective losses from multiple interfaces. We also overestimate the incident power by neglecting power loss due to Fresnel reflections off the fused silica-air interface.

## Evaluation of atomic trapping potential

Due to their high intensity gradients, phase singularities are effective as optical traps. Dielectric particles with a refractive index lower than the surrounding medium, reflective particles, and absorptive particles can all be trapped in the dark minimum of a beam, such as that on the axis of a donut beam carrying orbital angular momentum[25,26]. For neutral atoms, depending on the sign of the detuning $\Delta = \omega - \omega_O$ between the optical trap field frequency $\omega$ and a strong atomic resonance frequency $\omega_0$, such atoms are attracted to either intensity maxima (red $\Delta < 0$ detuning) or minima (blue $\Delta > 0$ detuning)[27]. Most optical dipole traps for neutral atoms are red traps which trap neutral atoms in arrays of tightly focused spots of light. Blue bottle traps with

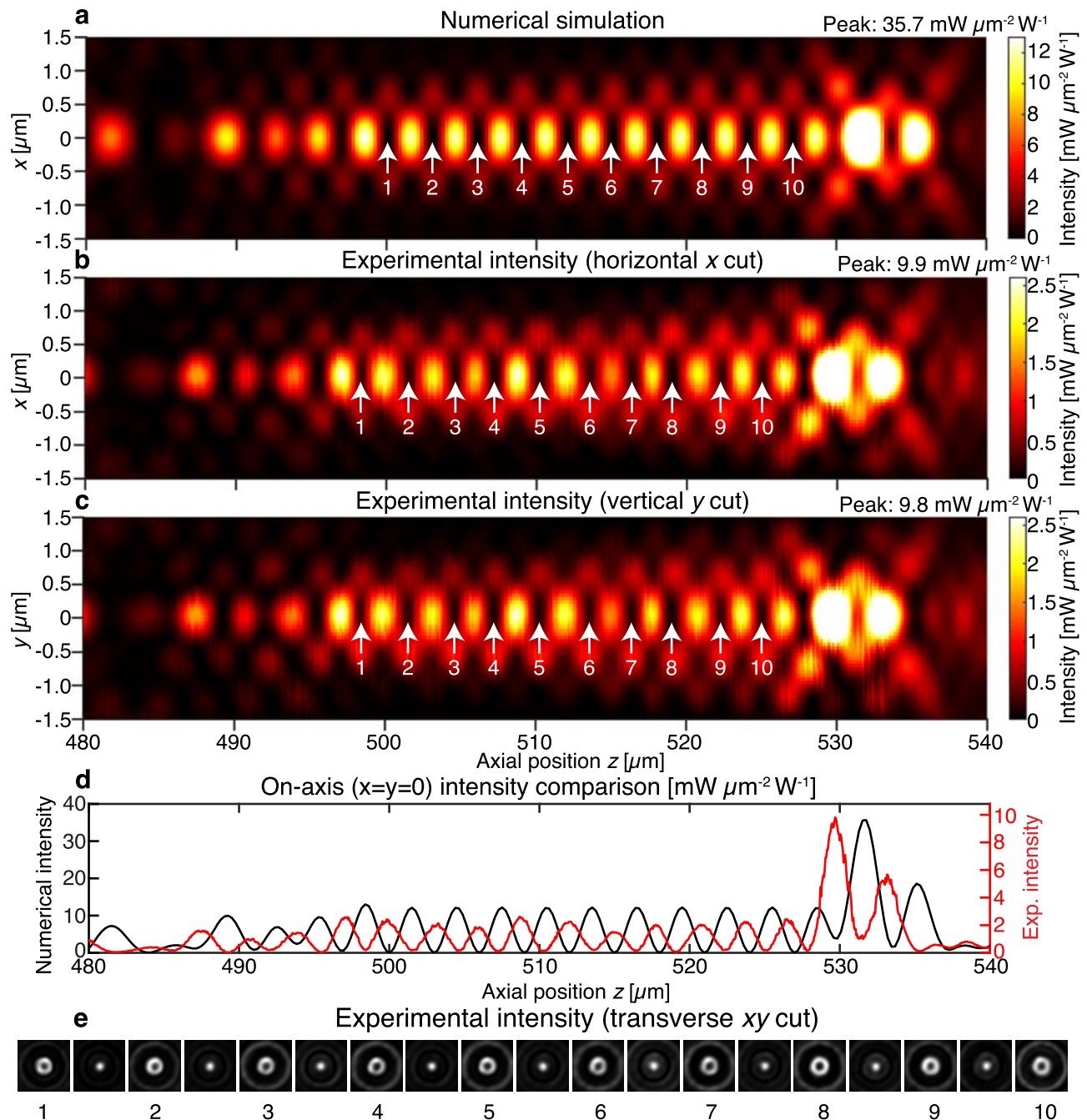

**Fig. 4 | Longitudinal intensity cuts for singularity array with ten on-axis 0D singularities.** The metasurface that produces this light field is located at z = 0. The color scales are adjusted to show the singular region with higher contrast; peak intensity values for each of the colormaps are indicates in the top right-hand corner. White arrows indicate the locations of the ten 0D singularities. **a** Numerically simulated xz cut for the ideal metasurface. The yz cut is identical due to the rotational symmetry of the light field about the optic axis. **b** Experimental xz cut and (**c**) experimental yz cuts for the fabricated metasurface light field,

demonstrating good agreement to the simulated light field. **d** On-axis (x = y = 0) intensity comparison for numerical (black) and experimental (red) measurements. The axial displacement arises due to the experimental illumination wavelength of 760.9 nm being slightly longer than the target wavelength of 760 nm. **e** Experimental transverse (xy) intensity profiles at each of the singularity positions (bright hollow annulus surrounding the 0D singularity) and in-between the singularity positions (focused spot). The longitudinal cuts in (**b**, **c**) are obtained by stacking 1201 such transverse images.

3D spatial confinement, which trap the atoms in a dark spot surrounded by light, are more difficult to realize but provide several key advantages over red traps. Atoms trapped in blue traps experience substantially lower scattering rates[27] and thereby longer coherence times[28]. Importantly, the trap laser can remain on during laser excitation with other coherent sources[29].

Techniques using a single structured beam have been able to produce single blue traps[28,30–32], more exotic bottle traps based on acoustic[33,34] or ponderomotive[35] forces, and arrays of blue traps in the transverse plane[29,36]. The state-of-the-art blue trap array in active use is arguably the quantum gas microscope[37], which holographically projects a two-dimensional optical lattice into a vacuum cell, thereby achieving thousands of trap sites with individual optical access. A key challenge is ensuring that all traps have identical optical environments, including spatially-varying polarization states, as these would shift the magnetic sublevels of the electronic states[38].

There is growing interest in using metasurfaces for the generation of atom traps[39–41], where the multifunctional, compact metasurface can replace multiple conventional optics and may even be located within the vacuum chamber. Recently, Hsu et al. performed single atom trapping with a red detuned trap generated by a metalens inside the vacuum chamber[41].

The geometrical parameters of the 0D singularity array shown here are compatible with that of cold $^{87}$Rb Rydberg atom arrays[42] ($D_2$ line at 780.241 nm) and may conceivably be deployed in the orthogonal geometry portrayed in Supplementary Fig. 15b, where a single metalens and single-sided illumination can generate the multiple blue traps for optical interrogation in the transverse direction. This is in contrast to the in-line architecture of optical traps in which the trapping and optical interrogation is performed through the same high numerical aperture objective. The trapping depth (in mK temperature units) per incident laser power is predicted to be 1.9 mK W$^{-1}$ for the numerical simulation and 0.2 mK W$^{-1}$ for the experimental intensity profile (Supplementary Information section 4). Both intensity profiles do not have any escape channels. Supplementary Information section 5 evaluates the sensitivity of the structured optical field to changes in incident illumination tilt and incident wavelength on the metasurface (the longitudinal intensity profiles are visualized in Supplementary Movie 1 and 2). Although the light field is tolerant to changes in the incident wavelength on the order of 10 nm, the effective angular bandwidth is around 2 mrad (0.11°). This is similar to the field of view of 0.2° obtained in the previously reported metasurface red optical tweezer with NA = 0.55[41]. This limited angular bandwidth may be overcome with metasurface angular dispersion engineering[43] to obtain better angular performance by trading off unneeded chromatic bandwith[44]. Although the cross-polarized $E_{y,z}$ fields were not explicitly controlled in the metasurface design process, the cross-sectional 3D polarization profile is highly similar across trap positions (Supplementary Fig. 7), producing similar optical environments and hence equal magnetic sublevel shifts for trapped atoms.

Passive metasurfaces excel in applications which afford very little volumetric and mass footprint while demanding high performance under a narrow set of constraints. The latter is due to the inherent trade-off between chromatic control, angular dispersion, and efficiency[44,45]. Given the space limitations in ultra-high-vacuum chambers and well-defined operational wavelengths for controlling and interrogating trapped particles in atomic physics, metasurfaces may be ideal for compact, few-component atom trap architectures. The 0D singularities generated by such metasurfaces are suitable for deployment as blue-detuned trap arrays and can also be accentuated in future work with dispersion engineering[46] to perform additional functions under illumination with different laser wavelengths or capture fluorescent emissions from the trapped atoms. Beyond optical traps, engineered 0D singularities may also be used in MINFLUX super-resolution microscopy[47] to capture information simultaneously from multiple points.

## Methods

### Computational design of metalens
The cylindrically symmetric phase-controlled metasurface at $z = 0$ mm is parametrized by a set of 1001 annular rings, each of 500 nm radial extent, to produce a total lens with a 500 µm radius. For each radial position, we assign a scalar $\phi$ for the propagation phase delay of light there. This treats the metasurface as phase-only and cylindrically-symmetric. We propagate this wavefront into the domain $z > 0$ using the vectorial diffraction integral[17] implemented on an automatically differentiable platform (Tensorflow[18]). For the first optimization stage, at each singularity position, we compute the $z$-directed phase gradient of the field $\partial\phi/\partial z$. The objective function $F_1$ to be minimized is the negative minimum of the squares of the $z$-directed phase derivatives

for each singularity position (Eq. 1).

$$F_1 = -\min\left\{\left(\frac{\partial\phi}{\partial z_i}\right)^2\right\}_{i=1,\ldots,n} \tag{1}$$

To improve convergence, we use a smooth approximation to the minimum function, which is analytic instead of being piecewise continuous (Eq. 2).

$$\min(a_1,\ldots,a_n) = \frac{\log\left[\sum_i \exp(-s \cdot a_i)\right]}{-s}, s = \frac{100}{\frac{1}{n}\sum_i|a_i|} > 0 \tag{2}$$

In the second optimization stage, the objective function $F_2$ to be minimized is the maximum of the deviations of the phase gradient to a large target phase gradient, set here to be 100 times the nominal field wavenumber $k_0$, plus penalty terms for differences in the second spatial derivative of the on-axis intensity $I(z) = |E_x(r = 0, z)|^2$ (Eq. 3).

$$F_2 = \max\left(\frac{\partial\phi}{\partial z_i} - 100k_0\right)^2_{i=1,\ldots,n} + c_1\frac{\sigma\left\{\partial_z^2 I(z_i)\right\}_{i=1,\ldots,n}}{\mu\left\{\partial_z^2 I(z_i)\right\}_{i=1,\ldots,n}} + c_2\frac{\sigma\left\{\partial_r^2 I(z_i)\right\}_{i=1,\ldots,n}}{\mu\left\{\partial_r^2 I(z_i)\right\}_{i=1,\ldots,n}}, \tag{3}$$

where $\sigma$ refers to the population standard deviation and $\mu$ is the population mean. $c_1$ and $c_2$ are weight parameters that are chosen so as to bring the three terms in $F_2$ onto similar scales. We use a smooth approximation to the maximum function (Eq. 4) to improve convergence, which is analogous to the smooth approximation to the minimum function described earlier.

$$\max(a_1,\ldots,a_n) = \frac{\log\left[\sum_i \exp(s \cdot a_i)\right]}{s}, s = \frac{100}{\frac{1}{n}\sum_i|a_i|} > 0, \tag{4}$$

Full optimization details are contained in Supplementary Information section 1.

### Nanofabrication of metalens
The metasurface comprises $TiO_2$ nanopillars on a glass substrate (0.5-mm-thick JGS2-fused silica) and is fabricated using electron beam lithography, atomic layer deposition, and reactive ion etching processes[23,48]. The nanopillar pattern is written into 700 nm thick ZEP520A electron-beam resist (Zeon Specialty Materials Inc.) using a high-speed 50 kV electron-beam lithography system (Elionix HS-50) followed by development in chilled o-Xylene (puriss. p.a., ≥ 99.0% (GC), Sigma Aldrich). The patterned holes are then conformally filled with amorphous $TiO_2$ through a low-temperature atomic layer deposition process (Cambridge NanoTech Savannah) until the holes are completely filled. The over-deposited $TiO_2$ is etched back using reactive ion etching with CHF$_3$/Ar/O$_2$ mixture (Oxford PlasmaPro 100 Cobra ICP Etching System) until the resist layer is exposed. The residual resist is removed by a downstream plasma asher (Matrix Plasma Asher, Matrix Systems Inc.), which leaves free-standing $TiO_2$ nanopillars. A 1.1 mm diameter opaque aperture is formed around the 1 mm diameter metasurface by photolithography using S1818 photoresist (Kayaku Advanced Materials Inc.), electron beam evaporation of 150 nm thick aluminum (Sharon electron beam evaporator), followed by a lift-off process via overnight immersion in Remover PG solution (Kayaku Advanced Materials Inc.).

### Experimental characterization of singularity array
A 760.9 nm single frequency distributed feedback (DFB) laser (TOPTICA Eagleyard GmbH) is driven with a constant current source (Newport 505 Laser Diode Driver) and kept at a constant temperature (Newport 325 Thermoelectric Cooler Driver). The single mode fiber-coupled output is collimated with a reflective collimator (Thorlabs RC12APC-P01) and is incident on the fused silica face of the

metasurface. The metasurface *z*-position is controlled using a closed-loop piezo-motor stage with nm resolution (Attocube ECSx3030). The transmitted light is captured using a horizontal microscope system comprising a high NA objective (Olympus 100x MPLAPON NA = 0.95), tube lens (Thorlabs TTL-180A) and CMOS camera (Thorlabs DCC1545M). The intensity image is captured over a range of longitudinal *z*-positions at steps of 50 nm, where *z* = 0 mm corresponds to the patterned surface of the metasurface. At each *z*-position, the system is allowed to stabilize for 10 s before multiple intensity images are captured at different exposure times ranging from 0.05 ms to 163 ms. These multiple exposure images are later weighted by their respective exposure times and stacked to remove saturated pixels and produce a composite image with a large intensity dynamic range.

Further experimental details are contained in Supplementary Information section 3.

### Reporting summary

Further information on research design is available in the Nature Portfolio Reporting Summary linked to this article.

## Data availability

The figure and supplementary data generated in this study have been deposited in the Figshare database under accession code https://doi.org/10.6084/m9.figshare.22580521[49].

## Code availability

The code that supports the findings of this study is available from the corresponding author upon request.

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

## Acknowledgements

S.W.D.L. is supported by A*STAR Singapore through the National Science Scholarship Scheme. J.S.P. is on leave from the Korea Institute of Science and Technology. M.L.M. is supported by NWO Rubicon Grant 019.173EN.010, by the Dutch Funding Agency NWO. This material is based upon work supported by the Air Force Office of Scientific Research under award number FA9550-22-1-0243 (F.C.). This work was performed in part at the Harvard University Center for Nanoscale Systems (CNS); a member of the National Nanotechnology Coordinated Infrastructure Network (NNCI), which is supported by the National Science Foundation under NSF award no. ECCS-2025158. The computations in this paper were run on the FASRC Cannon cluster supported by the FAS Division of Science Research Computing Group at Harvard University. The authors thank Mikhail Lukin (Harvard University), Brandon Grinkemeyer (Harvard University), and Sooshin Kim (Harvard University) for helpful discussions.

## Author contributions

S.W.D.L. conceived the algorithm and design with input from A.H.D. J.S.P. fabricated the samples. S.W.D.L., J.S.P., and D.K. designed the experiment and characterized samples. C.M.S. derived the connections to trap confinement and perturbation sensitivity. M.L.M. clarified applications of the system. S.W.D.L. wrote the manuscript with contributions from all authors. F.C. supervised the research.

## Competing interests

S.W.D.L., J.S.P., A.H.D, M.L.M., and F.C. are the inventors on a relevant provisional patent application (application number: US20230021549A1) owned by Harvard University. The authors declare no other competing interests.
