## [Peer Review File · Nature Communications]

Reviewers' Comments:

Reviewer #1:

Remarks to the Author:

In this work, the authors design and experimentally demonstrate a metasurface for creating a linear array of field minimums with applications such as creating 1D chains of optical traps for neutral atoms. They present a new optimization approach for obtaining these field minimums using inverse design. The experimental results are impressive and demonstrate great control over design and fabrication. The manuscript is well thought out and well written. The authors results are interesting, well presented, and appropriate for nature communications.

1. On pages 2 and 4, the authors state that 0D point singularities are uncommon because they are not topologically protected. What do the authors mean by topological protection? From the introduction, we are talking about topology from polarization singularities. If this is the case, 0D singularities can exist from the collision of two C-points along a 3D trajectory at off-axis points to create a 0D V-point singularity (with topological charge). [I.Freund, "Polarization singularity indices in Gaussian laser beams," Optics Communications, vol. 201, no. 4, pp. 251–270, Jan. 2002, doi: 10.1016/S0030-4018(01)01725-4.]
2. Going further along these lines, if the 0D singularities that are created on the axis are not protected by topology are they true singularities? Would these just be field minimums that are almost singularities that occur from optimization where the $|E|$ nearly goes to, but does not reach 0? Do the 2D cross sections of polarization ellipses for something like step 2, position 1 show a polarization singularity?

Reviewer #2:

Remarks to the Author:

The authors present the generation of multiple, point-like zeros in a linearly polarised, cylindrically symmetric field by shining a laser through an optimised metasurface. The manuscript is well written and is accompanied by very nice experimental results and extensive supplementary information. The metasurface optimisation technique itself is impressive and I do not doubt its potential for light manipulation with so many design degrees of freedom.

That said, I am concerned about the novelty of the work. The technique for generating the singularities is not new – the authors have already published work showing their metasurface optimisation methods for designing singularities (<https://doi.org/10.1038/s41467-021-24493-y>, which actually eludes to the point singularity), so any novelty in this work really lies in the type of singularity created – the 0D point singularity. Generically, scalar field zeros are infinite or closed strands in 3D space. These 'point singularities' are instead confined in all three dimensions, getting brighter in all outward directions, which as the authors say, certainly could be useful in spectroscopy techniques and optical trapping.

As the authors mention, point singularities have been previously demonstrated in vector fields by a crossing of multiple scalar field singularity strands (the zero-lines in each field component themselves are not point-like, but when they cross at a point the total field intensity, incorporating all spatial components, has a point-like zero), the type of singularity presented here is a different, point-like scalar field zero, which has unusual implications for OAM (i.e. there isn't any) – and this in particular is what I find interesting about this work. At a point singularity in the scalar field E , the $\text{Re}\{E\}=0$ and $\text{Im}\{E\}=0$ surfaces only touch tangentially so, interestingly, there is no circulation of phase around the singularity (of course, since the singularity is point-like, it would not be straightforward to define its topological charge, but intuitively this lack of phase circulation would imply $l=0$). Since singularities are widely associated with OAM, I am surprised that the authors did not mention this property of their

singularities.

Based on this, I do find this work to be interesting and I appreciate the sound discussion and quality of the experimental results. On the other hand, on the applications which the authors suggest, such as a blue-detuned optical trap, I do not think a point-like scalar field zero is more useful or than a point-like intensity zero in an electric field with arbitrary polarisation components, as have been published previously.

Overall, I do not think that the interesting topology of the singularity alone gives the work enough novelty for publication in Nature Communications, given that the authors' phase gradient optimisation technique, which makes up a large portion of the work, has already been published.

Reviewer #3:

Remarks to the Author:

The following comments should be addressed by the authors:

(i) Some of the ideas of the current submission has been proposed in the authors' previous publication [3]. Please emphasize the novelty in the current submission beyond what have been presented in [3].

(ii) In Fig. 2, the authors compare two methods of producing 0D singularity. The intensity minimization method only considers the intensity minimization at $z=0$ in the objective function. In general, for intensity optimization, a mask with desired intensity distribution will be defined in the optimization process to minimize the difference between the mask and the achievable field from the metasurface. As such, the nearby field distributions around the singularity can also be taken into account, and the broad intensity minimum problem in Fig. 2a can be resolved. Therefore, comparing the two methods, where a mask is used for intensity minimization is much fair. Moreover, intensity minimization is a straightforward approach because one has to check the intensity distribution after the phase gradient maximization. So why not directly optimize the intensity distribution?

(iii) In addition to the largest phase gradient, the intensity gradient is also the largest at the singularity. It will be interesting to investigate the intensity gradient maximization approach for the metasurface design.

(iv) Since some plots are saturated in Fig. 4, providing the normalized intensity distribution is suggested. A 1-D cut of the intensity distribution along the z -axis can also be added to show the singularities clearly.

Reviewer #1 (Remarks to the Author):

In this work, the authors design and experimentally demonstrate a metasurface for creating a linear array of field minimums with applications such as creating 1D chains of optical traps for neutral atoms. They present a new optimization approach for obtaining these field minimums using inverse design. The experimental results are impressive and demonstrate great control over design and fabrication. The manuscript is well thought out and well written. The authors results are interesting, well presented, and appropriate for nature communications.

1. On pages 2 and 4, the authors state that 0D point singularities are uncommon because they are not topologically protected. What do the authors mean by topological protection? From the introduction, we are talking about topology from polarization singularities. If this is the case, 0D singularities can exist from the collision of two C-points along a 3D trajectory at off-axis points to create a 0D V-point singularity (with topological charge). [I.Freund, "Polarization singularity indices in Gaussian laser beams," *Optics Communications*, vol. 201, no. 4, pp. 251–270, Jan. 2002, doi: 10.1016/S0030-4018(01)01725-4.]

We thank the reviewer for the positive comments. "Topological protection" refers to the robustness of a system against perturbations or defects, provided certain topological characteristics are preserved. For example, symmetry-protected bound states in the continuum are robust and continue to exist under small changes in system parameters, because of conserved and quantized topological charges that can only change under large system parameter deviations (Zhen et al., "Topological nature of optical bound states in the continuum", *Phys. Rev. Lett.* **113**, 25, 2014). In singular optics, topological protection refers to the persistent existence of the singularity under small changes in the surrounding medium or wave properties. They are referred to as elementary optical singularities and include, for instance, the canonical orbital-angular-momentum (OAM) singularity in a scalar field or bright C-points in polarization fields (Liu et al., "Topological polarization singularities in metaphotonics", *Nanophotonics* **10**, 5, 2021). Upon perturbation, such elementary optical singularities are merely displaced and can only be annihilated by a large perturbation that merges two singularities of opposite charge. Singularities that are not topologically protected, such as higher-order OAM modes, do not have this guarantee; they are destroyed or split into multiple topologically-protected singularities under field perturbation. This fragility to perturbation renders such singularities uncommon in nature. The topic of topological protection in optical singularities is thoroughly explored in the preprint cited: Spaegele et al., "Topologically protected four-dimensional optical singularities," ArXiv 2208.09054 (ref 8).

To better elucidate topological protection, we deferred its discussion from the introduction to the section "Geometry of 0D singularities", and have added the above paragraph to page 4 of the manuscript.

Not all topological charges lead to topological protection. A V-point is indeed a 0D singularity since it is an isolated zero field intensity position in a linearly polarized field. However, as the reviewer mentioned correctly, V-points are an edge case of two C-points colliding at a single point. This collision is not topologically protected, but is split back into the two topologically protected elementary C-points under arbitrarily small perturbation (Liu et al "Topological polarization singularities in metaphotonics", *Nanophotonics* **10**, 5, 2021).

Our 0D singularities are not topologically protected but are "true zero index vector singularities" with a Poincare-Hopf index of zero, to use the notation of Freund ("Polarization flowers", *Opt. Commun.* **199**, 2001). While this singularity array was designed for a scalar field and polarization-insensitive operation, the tight focusing rotates the incident field at the metasurface so as to produce small polarization components in the longitudinal and other transverse direction. To better display this behavior, we introduced two new supplementary figures to display the complex Cartesian fields (E_x , E_y , and E_z) and

polarization states at each of the ten transverse planes containing the 0D singularities (**Supplementary Fig. 7-8**), and added descriptive text to pages 8-9 of the manuscript. As exhibited by **Supplementary Fig. 7**, the transverse polarization near the singularities is dominated by linear x -polarization. Due to symmetry, the y and z polarization components vanish along the optic axis, and the total polarization state is undefined at the singularity positions. The 0D singularities are thus surrounded by linear polarization (**Supplementary Fig. 8**) with a constant azimuth of zero (oriented along the x -axis). This constant azimuth around the singular position implies a Poincare-Hopf index of zero.

Supplementary Figure 7. Cross-sectional xy cuts for the Cartesian field components (E_x , E_y , E_z) after the second optimization step. The electric field values are normalized to E_0 , the incident x -polarized electric field magnitude at the metasurface. Rows from top to bottom: Magnitude of E_x , phase of E_x , magnitude of E_y , phase of E_y , magnitude of E_z , phase of E_z . The plots are centered on ($x=0$, $y=0$).

Supplementary Figure 8. Cross-sectional xy cuts of the polarization azimuth (surface plot) and polarization ellipses (superimposed) after the second optimization step. The metasurface is illuminated with x -polarized light. Black ellipses indicate right elliptical/circular polarization and white ellipses indicate left elliptical/circular polarization. The plots are centered on $(x=0, y=0)$.

2. Going further along these lines, if the 0D singularities that are created on the axis are not protected by topology are they true singularities? Would these just be field minimums that are almost singularities that occur from optimization where the $|E|$ nearly goes to, but does not reach 0? Do the 2D cross sections of polarization ellipses for something like step 2, position 1 show a polarization singularity?

As mentioned previously, topological protection is not a necessary condition for the existence of a true singularity. For example, V-points are true 0D singularities where $|E|$ goes to zero, even though they are not topologically protected. Other singular structures, such as high order OAM modes which fracture into stable unity charge OAM components upon perturbation, also have $|E|=0$ along the optic axis.

We described our structures as 0D singularities since, in experimental realizations, there is no significant distinction between a mathematical singularity with identically zero $|E|$ and a field minimum with a non-measurable intensity below the instrument noise level. Our method thus enables the generation of arrays of dark spots that are asymptotic to the ideal singularity behavior subject to realization constraints. That being said, the reviewer notes correctly that in **Supplementary Fig 8** and **9** of the submitted manuscript, which

plots the zoomed-in field profile after including the non-uniform transmission amplitudes of the TiO_2 nanopillar library, multiple positions lose the crossing between the real and imaginary zero-isolines. These positions are not, strictly mathematically speaking, optical singularities since the field does not reach zero. This deviation arises because the 0D singularities are not topologically protected and were constructed by finely-tuning the metasurface phase profile under the assumption of ideal uniform transmission. Non-idealities arising from a realistic nanopillar library thus slightly distort the dark regions so that these positions are not mathematical singularities. While this deviation is likely not significant enough to impact applications which are sensitive to intensity profiles, closer field behavior to the mathematical ideal can be obtained by including non-uniform metasurface transmission during inverse design optimization, such as described in Lim et al, “A High Aspect Ratio Inverse-Designed Holey Metalens”, *Nano Lett.* **21**, 20, 2021. This discussion has been added to page 10 of the manuscript.

As discussed earlier, the 2D cross-sectional polarization plot in **Supplementary Fig. 7** in conjunction with the transverse field structure in **Supplementary Fig. 8**, demonstrate that the singular positions after Step 2 (without including non-uniform transmission from a TiO_2 nanopillar library) are “true zero index vector singularities” (V-points with zero Poincare-Hopf index in Freund’s terminology).

Reviewer #2 (Remarks to the Author):

The authors present the generation of multiple, point-like zeros in a linearly polarised, cylindrically symmetric field by shining a laser through an optimised metasurface. The manuscript is well written and is accompanied by very nice experimental results and extensive supplementary information. The metasurface optimisation technique itself is impressive and I do not doubt its potential for light manipulation with so many design degrees of freedom.

That said, I am concerned about the novelty of the work. The technique for generating the singularities is not new – the authors have already published work showing their metasurface optimisation methods for designing singularities (<https://doi.org/10.1038/s41467-021-24493-y>, which actually eludes to the point singularity), so any novelty in this work really lies in the type of singularity created – the 0D point singularity. Generically, scalar field zeros are infinite or closed strands in 3D space. These ‘point singularities’ are instead confined in all three dimensions, getting brighter in all outward directions, which as the authors say, certainly could be useful in spectroscopy techniques and optical trapping.

We appreciate the reviewer's concerns regarding the novelty of the work. We would like to clarify that the generation of closely-spaced, identical 0D singularities is indeed a non-trivial advancement, and the conclusions presented in our current study cannot be simply extrapolated from our earlier publication (ref [3]). Our aim is not to present our previously reported optimization technique as a novel concept in this work. Rather, as the reviewer correctly pointed out, we demonstrate the general applicability of our phase gradient optimization technique in creating singularities of an unexplored type (i.e., 0D) in any arbitrary arrangement in 3D (along the optical path). Our previous study which primarily focused on engineering singularities in a transverse (2D) plane.

Moreover, we would like to highlight that phase-gradient maximization alone is not sufficient for synthesizing an array of uniform point-singularities as traps, as demonstrated in this work. To achieve this, we also incorporated a third-order derivative in the objective function to produce the degenerate singularities, representing a significant development from our previous work. From an application standpoint, point singularities represent a fundamentally new class of structured light which can be favored in optical trapping, sensing, and microscopy. We provided an extensive analysis by quantifying the singularity performance and sensitivity in a practical trapping application. In light of these advancements,

we firmly believe that the work presented here represents a substantial contribution to the existing body of literature.

As the authors mention, point singularities have been previously demonstrated in vector fields by a crossing of multiple scalar field singularity strands (the zero-lines in each field component themselves are not point-like, but when they cross at a point the total field intensity, incorporating all spatial components, has a point-like zero), the type of singularity presented here is a different, point-like scalar field zero, which has unusual implications for OAM (i.e. there isn't any) – and this in particular is what I find interesting about this work. At a point singularity in the scalar field E , the $\text{Re}\{E\}=0$ and $\text{Im}\{E\}=0$ surfaces only touch tangentially so, interestingly, there is no circulation of phase around the singularity (of course, since the singularity is point-like, it would not be straightforward to define its topological charge, but intuitively this lack of phase circulation would imply $l=0$). Since singularities are widely associated with OAM, I am surprised that the authors did not mention this property of their singularities.

While it is true that literature makes an implicit equivalence between optical singularities and OAM, they are mutually independent in complex beams comprising superpositions of many eigenmodes (Berry and Liu, “No general relation between phase vortices and orbital angular momentum”, *Phys. A: Math. Theor.* **55**, 374001, 2022). Nevertheless, we find the reviewer's suggestion to explicitly discuss the lack of OAM behavior in this azimuthally-symmetric system useful and have plotted the transverse distribution of OAM density and energy flow (parametrized by the transverse projection of the Poynting vector) at each singularity plane as a new supplementary figure (**Supplementary Fig. 9**). These simulations confirm that the OAM contributions in the system are negligible.

We have added a paragraph discussing the lack of OAM in this system to page 9 of the manuscript.

Supplementary Figure 9. Transverse xy plot of numerically-simulated orbital angular momentum (OAM) density and power flux, after the second optimization step, at each singularity plane. The metasurface is illuminated with x -polarized light. The surface plot is the orbital angular momentum per incident power at the metasurface and the vector plot is the transverse projection of the Poynting vector. The plots are centered on the optic axis ($x=0, y=0$). The tiny OAM densities and lack of azimuthal circulation about the optic axis demonstrate that the OAM contributions are negligible in this system.

Based on this, I do find this work to be interesting and I appreciate the sound discussion and quality of the experimental results. On the other hand, on the applications which the authors suggest, such as a blue-detuned optical trap, I do not think a point-like scalar field zero is more useful or than a point-like intensity zero in an electric field with arbitrary polarisation components, as have been published previously.

While we have elected to demonstrate our technique with polarization-insensitive rotationally-symmetric nanostructures, the dynamics of which can be accounted for using a complex scalar field, we have already shown that singularities can also be engineered in vectorial fields with arbitrary transverse polarization (ref [3]). Furthermore, unlike the optical trapping of macroscopic particles, polarization effects exert a strong effect in atomic traps. In particular, improperly compensated polarization variations across space have a marked impact on atomic coherence and impair cooling (Thompson et al, “Coherence and Raman Sideband Cooling of a Single Atom in an Optical Tweezer”, *Phys. Rev. Lett.* **110**, 133001, 2013). It is also important that the optical environment (which includes polarization) is identical for all trapped atoms as they shift the

magnetic sublevels of electronic states (Garcia et al, “Improving the lifetime in optical microtraps by using elliptically polarized dipole light”, *Phys. Rev. A* **97**, 023406, 2018)) and we have added this consideration to page 13 of the manuscript. It is thus essential for us to be able to control not only the intensity profile but also the spatially-varying phase and polarization maps across multiple distinct trap positions. This is easily done for single trap positions, but exerting this control for multiple tightly-spaced traps (to allow inter-atom interaction) is highly nontrivial and is the primary application of the work presented here.

Overall, I do not think that the interesting topology of the singularity alone gives the work enough novelty for publication in Nature Communications, given that the authors’ phase gradient optimisation technique, which makes up a large portion of the work, has already been published.

We hope that the description of novelty in the previous paragraphs have addressed the reviewer’s concerns.

Reviewer #3 (Remarks to the Author):

The following comments should be addressed by the authors:

(i) Some of the ideas of the current submission has been proposed in the authors' previous publication [3]. Please emphasize the novelty in the current submission beyond what have been presented in [3].

We hope that the description of novelty in the Reviewer 2 response has addressed the reviewer’s concerns.

(ii) In Fig. 2, the authors compare two methods of producing 0D singularity. The intensity minimization method only considers the intensity minimization at $z=0$ in the objective function. In general, for intensity optimization, a mask with desired intensity distribution will be defined in the optimization process to minimize the difference between the mask and the achievable field from the metasurface. As such, the nearby field distributions around the singularity can also be taken into account, and the broad intensity minimum problem in Fig. 2a can be resolved. Therefore, comparing the two methods, where a mask is used for intensity minimization is much fair. Moreover, intensity minimization is a straightforward approach because one has to check the intensity distribution after the phase gradient maximization. So why not directly optimize the intensity distribution?

Conventional numerical techniques for light-field shaping work well for shaping lighted regions but perform poorly in shaping the low intensity regions, especially when these regions are comparable to the characteristic Airy disk size of the optics (near the highest spatial frequency that can be controlled). In computer-generated holography, for instance, reverse propagation from a singular region with zero intensity provides no information. Specifying one intensity pattern around singularities also excludes other potentially better-behaved fields and artificially limits the search space. The specified intensity structure may also not correspond to a physically realizable solution to Maxwell’s equations. We have discussed the relative merits of specifying an intensity distribution to phase gradient maximization in the manuscript and supplementary information of our previous publication (reference 3, Lim et al, “Engineering phase and polarization singularity sheets”, *Nat. Commun.*, **12**, 4190, 2021).

(iii) In addition to the largest phase gradient, the intensity gradient is also the largest at the singularity. It will be interesting to investigate the intensity gradient maximization approach for the metasurface design.

Although there are large intensity gradients in the vicinity of an optical singularity, they do not overlap. As the intensity is a non-negative number, its first spatial derivative (gradient) cannot be zero when the intensity is zero, as it would require negative intensity values.

That being said, we have investigated intensity gradient maximization for designing an array with the same geometry as the manuscript metasurface, added a supplementary figure summarizing the results of the optimization (**Supplementary Fig. 10**), and added a new **Supplementary Information section 2** for this discussion. The optimization procedure is identical with the exception of the objective function being written in terms of the z -directed intensity gradient instead of the z -directed phase gradient. **Supplementary Fig. 10a-b** exhibits the xz intensity profile after Step 1 optimization, in which the z -directed intensity gradients at ten equally-spaced on-axis positions from $z=500$ μm to $z=527$ μm are maximized. The tunable parameters are the same as in the manuscript device: the 1001 phase values on the metasurface radial profile. The resultant intensity and phase gradients along the optical axis ($x=y=0$) are also plotted in **Supplementary Fig. 10c**.

We notice that the dark regions are non-systematically displaced in the z -direction from the positions of intensity gradient maximization, and that the phase gradients in the adjacent dark regions are not appreciably larger in magnitude than the vacuum wavenumber k_0 . However, the absolute values of the intensity are substantially larger than those produced by phase gradient maximization due to most field structure being concentrated near the optimization points, whereas the device obtained through phase gradient maximization produces field structure further away from the optimization points, thereby spreading out the wave energy. Intensity maximization may be useful for applications which are less sensitive to the precise positioning of the dim regions or which emphasize contrast between the dark and bright regions.

Supplementary Figure 10. Array designed using z -directed intensity gradient maximization. **a** xz total intensity profile, white crosses are the positions at which the intensity gradient was maximized. **b** log-scaled xz total intensity profile. **c** Intensity and phase gradient profile along the optical axis ($x=y=0$). E_0 is the incident electric field at the metasurface and k_0 is the vacuum wavenumber. The vertical dotted lines are the positions at which the intensity gradient was maximized. The maximum phase gradient position and the minimum intensity positions are displaced in an inconsistent fashion from the peak intensity gradient positions.

(iv) Since some plots are saturated in Fig. 4, providing the normalized intensity distribution is suggested. A 1-D cut of the intensity distribution along the z -axis can also be added to show the singularities clearly.

We have added a 1D cut along the z -axis as Fig. 4d and the normalized intensity distribution without saturation as Supplementary Fig. 14. Since the emphasis of this manuscript is on the local behavior in the vicinity of the singularities, we have elected for a saturated intensity plot in Fig. 4a-c to better show the intensity profile around the relevant region: around the ten singularities.

Updated manuscript **Fig. 4** with the added on-axis intensity comparison in panel **d**.

Supplementary Figure 14. Longitudinal intensity profile comparison between the numerical simulation and experiment. These images are identical to that of **Fig. 4a-c** in the main text with the colorbar adjusted to show the full dynamic range of intensities without saturation.

Reviewers' Comments:

Reviewer #1:

Remarks to the Author:

The authors have responded to all of my questions and concerns resulting in changes to the manuscript. I can recommend it for publication. I am still weary of referring to the points as singularities, since a true singularity would result in V-points (S_0 Stokes parameter $\rightarrow 0$) leading to a polarization ellipse singularity that should show up with polarization winding in supplementary figure 8. The lack of such winding suggest to me that these are only near singularities.

Also for the authors knowledge, it should be noted that symmetry protected BICs are indeed V-points that can be split into C-points during certain symmetry breaking operations. The presence of a symmetry protected BIC can be thought of as two C-points colliding at γ . See W. Liu, B. Wang, Y. Zhang, et al., "Circularly polarized states spawning from bound states in the continuum," Phys. Rev. Lett., vol. 123, p. 116104, 2019.

Reviewer #2:

Remarks to the Author:

The authors have addressed my original concerns, primarily with novelty versus reference [3] in the manuscript. Although both works do use fundamentally similar techniques to generate the singularities, I appreciate the authors arguments on novelty and believe the work is suitable for Nature Communications. I am also happy that the authors have discussed OAM implications of the presented singularity.

I have a couple of minor comments the authors may wish to consider:

The authors laid out that polarisation effects are significant for atomic traps and that a point singularity in a scalar field ensures uniform spatial polarisation, and have made an addition on page 13 of the manuscript. Though I suggest the authors make a stronger distinction between the presented 0D scalar singularity from a vector singularity in terms of polarisation effects for atomic optical trap applications in the introduction, perhaps somewhere on page 3 (since it is in the introduction that vector singularities from the literature are discussed).

I am aware of the authors' other publication <https://arxiv.org/abs/2208.09054>, which discusses the stability of a singularity in terms of the mathematical conditions which are satisfied and the dimension of the parameter space. I wonder what this means for the presented scalar field 0D singularity, because only two scalar fields ($\text{Re}\{E\}$ and $\text{Im}\{E\}$) are zero, but presumably first order derivatives in the radial direction (since we have cylindrical symmetry) must also be zero to achieve the parabolic shape of the $\text{Re}\{E\} = 0$ and $\text{Im}\{E\} = 0$ surfaces. Perhaps the authors could comment on the stability of the singularity in this sense (how many mathematical conditions are required, versus the number of real space dimensions).

Reviewer #3:

Remarks to the Author:

This reviewer thanks the authors to provide thorough responses to the comments raised. They illustrated the advantages of using the phase gradient maximization over the intensity gradient maximization. Therefore, this submission is recommended for publication. However, there is a minor comment. In Fig. 4d, there exists a large discrepancy between the locations of the intensity peak and dip for the simulated and experimental results. Please provide a short discussion on cause of the discrepancy other than attributing this to fabrication tolerance of the metasurface.

Reviewer #1 (Remarks to the Author):

The authors have responded to all of my questions and concerns resulting in changes to the manuscript. I can recommend it for publication. I am still weary of referring to the points as singularities, since a true singularity would result in V-points (S_0 Stokes parameter $\rightarrow 0$) leading to a polarization ellipse singularity that should show up with polarization winding in supplementary figure 8. The lack of such winding suggests to me that these are only near singularities.

We appreciate the Reviewer's concern about the singularity nomenclature. We agree that some of these positions are only near singularities, and have discussed this in the manuscript, page 8: "In several positions, the real and imaginary zero-isolines come close but do not touch, indicating that these situations are close approximations of 0D singularities and not mathematical 0D singularities". Other singularities are much better approximations, like positions 5-10 of Supplementary Figure 5.

Also for the authors' knowledge, it should be noted that symmetry protected BICs are indeed V-points that can be split into C-points during certain symmetry breaking operations. The presence of a symmetry protected BIC can be thought of as two C-points colliding at Γ . See W. Liu, B. Wang, Y. Zhang, et al., "Circularly polarized states spawning from bound states in the continuum," *Phys. Rev. Lett.*, vol. 123, p. 116104, 2019.

Thank you for the information. We have added a reference to elaborate on this behavior in the manuscript, page 4: "Singularities that are not topologically protected, such as higher-order OAM modes, do not have this guarantee; they are destroyed or split into multiple topologically-protected singularities under field perturbation. *This is also observed in BICs when symmetry-breaking operations cause a V-point to fragment into topologically-protected C-points (Liu et al, Phys Rev Lett 123, 116104, 2019).* This fragility to perturbation renders such singularities uncommon in nature."

Reviewer #2 (Remarks to the Author):

The authors have addressed my original concerns, primarily with novelty versus reference [3] in the manuscript. Although both works do use fundamentally similar techniques to generate the singularities, I appreciate the authors' arguments on novelty and believe the work is suitable for Nature Communications. I am also happy that the authors have discussed OAM implications of the presented singularity.

I have a couple of minor comments the authors may wish to consider:

The authors laid out that polarisation effects are significant for atomic traps and that a point singularity in a scalar field ensures uniform spatial polarisation, and have made an addition on page 13 of the manuscript. Though I suggest the authors make a stronger distinction between the presented 0D scalar singularity from a vector singularity in terms of polarisation effects for atomic optical trap applications in the introduction, perhaps somewhere on page 3 (since it is in the introduction that vector singularities from the literature are discussed).

Realizing atom traps with fixed polarization ensures uniform magnetic sublevel shifts for the trapped atoms. To discuss this we have added a sentence to the introduction, page 4: “While the singularities are engineered for a scalar field corresponding to a fixed linear polarization, we also examine the full 3D polarization distribution that would be generated by a realistic wavefront shaping device like a metasurface.” and a sentence to the results section, page 14 “Although the cross-polarized $E_{y,z}$ fields were not explicitly controlled in the metasurface design process, the cross-sectional 3D polarization profile is highly similar across trap positions (Supplementary Figure 7), producing similar optical environments and hence equal magnetic sublevel shifts for trapped atoms.”

I am aware of the authors’ other publication <https://arxiv.org/abs/2208.09054>, which discusses the stability of a singularity in terms of the mathematical conditions which are satisfied and the dimension of the parameter space. I wonder what this means for the presented scalar field 0D singularity, because only two scalar fields ($\text{Re}\{E\}$ and $\text{Im}\{E\}$) are zero, but presumably first order derivatives in the radial direction (since we have cylindrical symmetry) must also be zero to achieve the parabolic shape of the $\text{Re}\{E\} = 0$ and $\text{Im}\{E\} = 0$ surfaces. Perhaps the authors could comment on the stability of the singularity in this sense (how many mathematical conditions are required, versus the number of real space dimensions).

In contrast to the study mentioned in the preprint (Topologically protected four-dimensional optical singularities by C. Spaegele *et al*), the point singularities in this manuscript do not exhibit topological protection and hence are not stable to field perturbations, as discussed in the introduction, page 4. Nevertheless, for the atomic trapping applications discussed, external field perturbations can be kept negligibly low by appropriate shielding, making topological stability less of a concern for practical experimental deployment.

Reviewer #3 (Remarks to the Author):

This reviewer thanks the authors to provide thorough responses to the comments raised. They illustrated the advantages of using the phase gradient maximization over the intensity gradient maximization. Therefore, this submission is recommended for publication. However, there is a minor comment. In Fig. 4d, there exists a large discrepancy between the locations of the intensity peak and dip for the simulated and experimental results. Please provide a short discussion on cause of the discrepancy other than attributing this to fabrication tolerance of the metasurface.

The axial displacement arises due to the illumination wavelength of 760.9 nm being slightly longer than the target wavelength of 760 nm. We have added the following sentence to the main text (page 12) and the Fig 4d caption: “The axial displacement of the experimental intensity profile with respect to the simulated profile in the negative z-direction can be attributed to the incident laser wavelength of 760.9 nm being slightly longer than the design wavelength of 760 nm.”